# A systematic review and meta-analysis quantifying schistosomiasis infection burden in pre-school aged children (PreSAC) in sub-Saharan Africa for the period 2000–2020

Chester Kalinda[1,2]*, Tafadzwa Mindu[2], Moses John Chimbari[2]

1 University of Namibia, Katima Mulilo, Namibia, 2 Department of Public Health, College of Health Sciences, School of Nursing and Public Health, University of KwaZulu-Natal, Durban, South Africa

* ckalinda@gmail.com, ckalinda@unam.na, KalindaC@ukzn.ac.za

## Abstract

### Introduction

Following the adoption of the World Health Assembly Resolution WHA 65.21 and Neglected Tropical Diseases road map 2021–2030, schistosomiasis control programmes have shifted from morbidity control to disease elimination. However, several gaps continue to be observed in the implementation of control programmes with certain age groups omitted from these campaigns increasing health inequalities and risks of reinfections to previously treated groups. We used the Inverse Variance Heterogeneity (IVhet) model to estimate the prevalence of schistosomiasis infection among preschool-aged children.

### Methods

We did a systematic review of peer-reviewed literature on schistosomiasis in sub-Saharan Africa for the period January 1, 2000 to November 30, 2020. Quantitative data for cases of schistosomiasis infection were extracted, including country and region where the studies were done, year of publication and specific schistosome species observed. The IVhet model was used to estimate the pooled prevalence estimate (PPE), the heterogeneity and publication bias.

### Results

We screened 2601 articles to obtain 47 eligible studies containing quantitative data on pre-school-aged children. Of the selected studies, 44.7% (n = 22) were from East Africa while the least number of studies obtained (2.1%, n = 1) was from Central Africa. 21712 subjects were screened for infection due to *Schistosoma* spp; 13924 for *S. mansoni* and 7788 for *S. haematobium*. The PPE for schistosomiasis among PreSAC was 19% (95% CI: 11–28). Infection due to *S. mansoni* (IVhet PPE: 22% (95% CI: 9–36) was higher than that due to *S. haematobium* (15%; 95% CI: 6–25). A Luis Furuya–Kanamori index of 1.83 indicated a lack of publication bias. High level of heterogeneity was observed (I2 > 90%) and this could not be reduced through subgroup analysis.

**Data Availability Statement:** All data are within the paper and its Supporting information files.

**Funding:** This study was not funded by any institution.

**Competing interests:** The authors have declared that no competing interests exist.

## Conclusion

Schistosomiasis infection among pre-school aged children 6 years old and below is high. This indicates the importance of including this age group in treatment programmes to reduce infection prevalence and long-term morbidities associated with prolonged schistosome infection.

## Introduction

Schistosomiasis is one of the most common parasitic infections caused by blood fluke trematodes and mainly affects poor and marginalized communities with inadequate sanitation and health services [1]. In terms of the number of people affected and at risk of infection, schistosomiasis has been ranked second to malaria [2]. Infection with *Schistosoma* parasites reduces household income and economic productivity [3] and if left untreated, long term morbidity may occur [4–6]. Control of schistosomiasis is a fundamental part of the Sustainable Development Goal (SDG) 3 which seeks to ensure healthy lives and promote well being for all ages with the health target 3.3 in part focusing on ending neglected tropical diseases by 2030, thus contributing to the achievement of universal health coverage (Health target 3.8). Achieving these targets in developing countries is dependent on supporting the research and development of vaccines and medicines, early warning, risk reduction and management of health risks (Target 3b) [7, 8].

Efforts to control schistosomiasis are ongoing through various global and national disease monitoring and control initiatives [9, 10]. These programmes have often been based on school mass drug administration (MDA) using praziquantel. The success indicator has been based on treatment coverage of 75% among school-aged children (SAC) [11, 12] with the inclusion of adults in high-risk settings where disease prevalence in SAC is 50% [12]. Schistosomiasis infection among SAC has often been used as a proxy to determine endemicity and forms the basis for disease control [12, 13] and this has been justified by several school-based studies that observed high prevalence and intensity rates among enrolled SAC. Furthermore, the use of school-based schistosomiasis control programmes in several settings has been cost-effective because of availability ofalready existing infrastructure [14].

However, the success of school-based MDA programmes is highly dependent on the number of SAC enrolled [15] and this usually excludes unenrolled children, those absent from school at the time of implementation [16], preschool-aged children (PreSAC), and certain adolescents and adults [17]. Although there is evidence of schistosomiasis infection among PreSAC, this age group has not been included in many MDA programmes [17]. This increases health inequalities and accumulation of potentially irreversible morbidities due to prolonged infection that ultimately compromises the future wellbeing of children that are not treated early [18]. To design cost-effective monitoring and control activities for schistosomiasis, determination of infection prevalence even among PreSAC is a critical component. We, therefore, used the inverse variance heterogeneity (IVhet) model to estimate the prevalence of schistosomiasis infection among PreSAC in sub-Saharan Africathus illuminate the importance of their inclusion in MDA programmes.

## Methods

### Search strategy and selection criteria

We searched for literature using the Preferred Reporting Items for Systematic Reviews and Meta-Analyses (PRISMA) guidelines [19] in PubMed, MEDLINE, EMBASE and Web of

Science databases. Literature searched was based on paediatric schistosomiasis published between January 1, 2000, and November 30, 2020. A broad search strategy combining several terms and restricted to SSA was used. The current study used the search terms "schistosomiasis AND pre-school" OR "pre-school" OR "under five" AND "sub-Saharan Africa" NOT "school-age children" (S1 File). We included studies that were published as peer-reviewed journals, reports and book chapters. In addition, we identified more relevant articles from reference lists of already identified articles. Two reviewers (TM and CK) independently screened through the titles and abstracts to identify relevant reports. Any disagreements were resolved by discussion unless otherwise arbitrated by the third author (MJC). The inclusion criteria for all articles were:

i. Articles reporting data from any SSA country

ii. Articles reporting prevalence rates among PreSAC

iii. Articles that included data on the following outcome of interest: specific schistosome studied, sample size and number of positive cases, age groups of respondents.

iv. Articles reporting the age of participants to be between 0–5 years old.

## Data abstracted and quality appraisal

Data extracted from the reviewed papers included the first author's name, year of publication, study country, the region in Africa, sample size, number of cases and age range. The quality of all studies included was assessed using the Joanna Briggs Institute Prevalence Critical Appraisal Tool [20]. Each selected study was assessed using 10 quality control items and for each item fulfilled, a score of 1 was given while a 0 was given for each unfulfilled item. An aggregate of all the scores was generated and converted into an index. Based on the quality indices generated, studies were classified as having low (0.0–0.3), moderate (0.4–0.6) or high (0.7–1.0) quality (S2 File).

## Data analysis

We used the inverse variance heterogeneity (IVhet) model [21] in MetaXL to obtain the Pooled prevalence estimates (PPE) for the selected studies. Compared to the fixed effect (FE) or random effect (RE) models, the IVhet model regardless of heterogeneity, maintains a correct coverage probability at a lower detected variance [21]. This ensures no underestimation of the statistical error and it maintains modest estimates compared to the RE model. The level of heterogeneity was evaluated using Cochran's Q statistic and $I^2$ while publication bias was assessed using the Luis Furuya–Kanamori (LFK) index of the Doi plot [22]. We determined the symmetry of the Doi plots using the LFK index. An LFK index shows the level of publication bias depending on the magnitude of the index. An LFK value in the range of '±1' was considered as 'symmetrical' and the level of bias classified as "absence of publication bias". On the other hand, an index value of '±2' was considered as minor asymmetry and classified as "low publication bias" while an index value outside the range of '±2' was classified as major asymmetry and "high publication bias" [22]. Forest plots were used to display the estimated prevalence and their 95% confidence interval. To explore heterogeneity and factors that could potentially influence the observed PPE, we implemented subgroup analysis by stratifying our data according to parasite species and the regions (West Africa, Central, East Africa or Southern Africa) where the studies were conducted; thus assessing heterogeneity between subgroup and within-group.

## Results

### Search results

Fig 1 summarizes the selection process that was used following the PRISMA guidelines. The initial search yielded 2601 studies. After removing duplicates and studies that were deemed irrelevant, following the set inclusion criteria, a total of 47 studies were selected for meta-analysis (Fig 1).

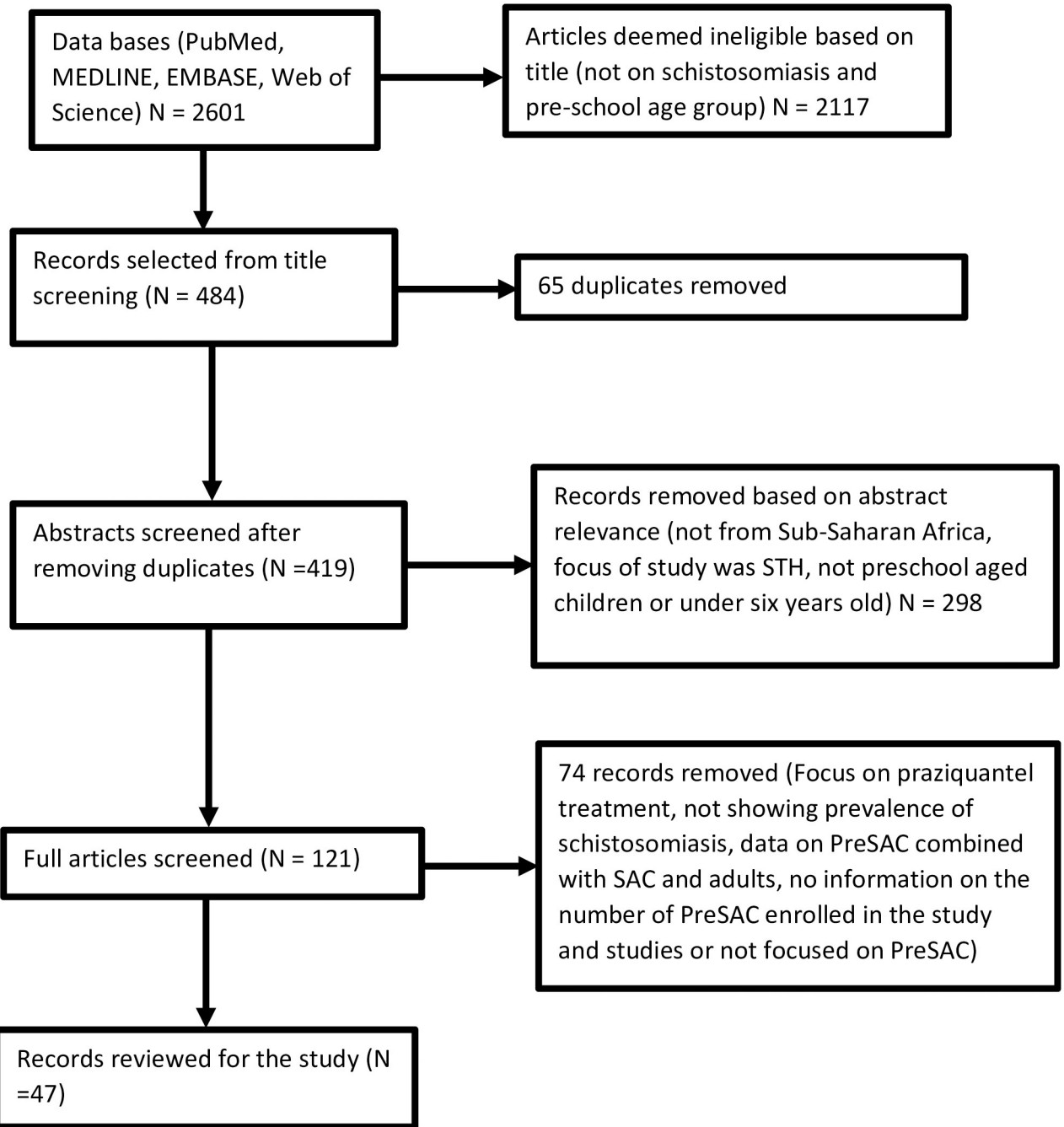

**Fig 1. PRISMA flow of study selection.**

## Study characteristics and PPE analysis

Forty-seven studies were eligible for final inclusion in the study. These studies were conducted in sixteen countries; 4% [23] were from Central Africa (Cameroon), 22% [24–33] were from Southern Africa (Angola, eSwatini, Malawi, South Africa and Zimbabwe), 31% [34–48] were from West Africa (Cote d'Ivoire, Ghana, Mali, Niger, Nigeria and Sierra Leone) and 43% [49–69] were from East Africa (Ethiopia, Kenya, Rwanda, Tanzania and Uganda). Furthermore, 64.1% of the subjects were screened for morbidity outcomes due to *S. mansoni* while 35.9% were screened for *S. haematobium*. Overall, the studies enrolled a total of 21712 PreSAC.

A wide range of diagnostic tests was used to detect the presence of schistosome ova in urine and stool samples (Table 1). Kato Katz was the commonly used diagnostic test for studies screening for *S. mansoni* infection (96.6%, n = 28/29) with one study using the Sodium acetate-acetic acid-formalin (SAF) solution concentration method alone. Other studies used a combination of Kato Katz with SAF (n = 3), Kato Katz and point-of-care test circulating cathodic antigen (POC-CCA) (n = 10), Kato Katz and enzyme-linked immunosorbent assays (ELISAs) (n = 4) and real-time polymerase chain reaction (qPCR) (n = 1). On the other hand, the urine filtration method was widely used among those studies screening for *S. haematobium* (95.6%, 22/23) while one study used micro-hematuria in urine measured using urine reagent strips as a proxy-diagnosis for *S. haematobium*. Some studies (n = 6) also used a combination of urine filtration methods and the dipstick assay for the diagnosis of *S. haematobium* (Table 1).

In the overall analysis, the IVhet model showed a pooled prevalence estimate (PPE) of 19% (95% CI: 11–28), with a high degree of heterogeneity (I2 = 99%, *p* < 0.001). Subgroup analysis stratified by schistosome species showed that infection due to *S. mansoni* was 22% (95% CI: 9–36) while that due to *S. haematobium* was 15% (95% CI: 6–25) (Fig 2).

Furthermore, stratification of infection by region showed that East Africa had a combined PPE of 27% (95% CI: 12–43) while only one study was obtained from Central Africa and had a PPE of 10% (95% CI: 0–27) (Fig 3). High level of heterogeneity was observed (I2 > 90%) and this could not be reduced through subgroup analysis by sub-region nor schistosome species. No significant publication bias was observed both from the funnel plot and doi plot as shown by the LFK index of 1.83 which indicates minor asymmetry (S1 and S2 Figs).

## Discussion

Our findings reinforce observations that have been made in several epidemiological surveys regarding the transmission of schistosomiasis among PreSAC. This contributes to the growing evidence that schistosomiasis infection in high endemicity settings cuts across all age groups, thus justifying the need to include PreSAC in disease monitoring and control programmes [18, 59, 70, 71]. Considering the operational difficulties associated with collection of samples for parasitology from PreSAC [18, 72], the prevalence of schistosomiasis among PreSAC may even be higher, a potential reflection of their continued exclusion from MDA programmes [17, 73]. Although studies examining the prevalence of schistosomiasis have been reporting infection among several other age groups for several years, global interests in determining and controlling infection among PreSAC has surged lately. Our study findings support the current global agenda of improving child health and advocates for the inclusion of PreSAC in regional and national mass drug administration with praziquantel for schistosomiasis control [15, 71].

We observed that the prevalence estimates for *S. mansoni* (IVhet PPE: 22% (95% CI: 9–36) were higher than for *S. haematobium* (IVhet PPE: 15%, 95% CI: 6–25). Contrary to our observations, an earlier study that used a Bayesian geostatistical modelling approach reported prevalence levels of 17·4% and 8% for *S haematobium* and *S mansoni*, respectively among SAC [74].

**Table 1. List of studies included and diagnostics tests used.**

| Author name and Year | Sample size | Positive cases | Schistosome species | Country | Regions | Technique | Sampling strategy |
|---|---|---|---|---|---|---|---|
| Macklina et al. (2018) | 21 | 2 | *S. haematobium* | Cameroon | Central Africa | Urine filtration | |
| Chimponda and Mduluza (2020) | 145 | 31 | *S. haematobium* | Zimbabwe | Southern Africa | Urine filtration | Simple random |
| Chu et al. (2010) | 59 | 9 | *S. haematobium* | ESwatini | Southern Africa | Urine filtration | Simple random |
| Moyo et al. (2016) | 143 | 19 | *S. haematobium* | Malawi | Southern Africa | Urine filtration | Simple random |
| Mutsaka-Makuvaza et al. (2018) | 535 | 71 | *S. haematobium* | Zimbabwe | Southern Africa | Urine filtration | Simple random |
| Osakunor et al. (2018) | 1502 | 128 | *S. haematobium* | Zimbabwe | Southern Africa | Urine filtration | Systematic sampling |
| Sousa-Figueiredo et al. (2012) | 1237 | 124 | *S. haematobium* | Angola | Southern Africa | Urine strips | Simple random |
| Mduluza-Jokonya et al. (2020) | 415 | 145 | *S. haematobium* | Zimbabwe | Southern Africa | Urine filtration | Simple random |
| Poole et al. (2014) | 208 | 103 | *S. mansoni* | Malawi | Southern Africa | Kato Katz, POC CCA, ELISA | Semi-random |
| Sacolo-Gwebu et al. (2019) [1] | 998 | 9 | *S. haematobium* | South Africa | Southern Africa | Urine filtration | Systematic random sampling |
| Sacolo-Gwebu et al. (2019) [2] | 1143 | 11 | *S. mansoni* | South Africa | Southern Africa | Kato Katz | Systematic random sampling |
| Wami et al. (2015) | 104 | 14 | *S. haematobium* | Zimbabwe | Southern Africa | Urine filtration, SEA ELISA, Reagent strips | Systematic random sampling |
| Armoo et al. (2020) | 190 | 48 | *S. mansoni* | Ghana | West Africa | urine-CCA, real-time PCR, Kato-Katz | Systematic random sampling |
| Garba et al. (2010) [1] | 185 | 81 | *S. mansoni* | Niger | West Africa | Kato-Katz | No sampling. All eligible participants included |
| Garba et al. (2010) [2] | 282 | 161 | *S. haematobium* | Niger | West Africa | Urine filtration | No sampling. All eligible participants included |
| Adeniran et al. (2017) [1] | 167 | 6 | *S. mansoni* | Nigeria | West Africa | Sodium acetate acetic-acid formalin solution | Not specified |
| Adeniran et al. (2017) [2] | 167 | 8 | *S. haematobium* | Nigeria | West Africa | Urine filtration, Reagent strip | Not specified |
| Babatunde et al. (2013) | 72 | 16 | *S. haematobium* | Nigeria | West Africa | Urine filtration | Random sampling |
| Bosompem et al. (2004) [1] | 80 | 9 | *S. haematobium* | Ghana | West Africa | ELISA, Urine filtration, Reagent strip | Not mentioned |
| Bosompem et al. (2004) [2] | 75 | 0 | *S. mansoni* | Ghana | West Africa | Kato Katz | Not mentioned |
| Coulibaly et al. (2013) [1] | 242 | 26 | *S. haematobium* | Ivory Coast | West Africa | Urine filtration, POC CCA | All eligible children included in the study |
| Coulibaly et al. (2013) [2] | 242 | 56 | *S. mansoni* | Ivory Coast | West Africa | Kato Katz | All eligible children included in the study |
| Dabo et al. (2011) | 338 | 173 | *S. haematobium* | Mali | West Africa | Urine filtration | No sampling was done due to the small population size |
| Ekpo et al. (2010) | 167 | 97 | *S. haematobium* | Nigeria | West Africa | Urine filtration | No sampling was done due to the small number of PSAC |
| Ekpo et al. (2012a) | 83 | 14 | *S. haematobium* | Nigeria | West Africa | Urine filtration | No sampling was done due to the small number of PSAC |
| Ekpo et al. (2012b) | 86 | 45 | *S. haematobium* | Nigeria | West Africa | Urine filtration, Urine dipstick | No sampling was done due to the small number of PSAC |
| Hodges et al. (2012) | 1803 | 202 | *S. mansoni* | Sierra leone | West Africa | Kato Katz | Random selection |

(*Continued*)

**Table 1.** (Continued)

| Author name and Year | Sample size | Positive cases | Schistosome species | Country | Regions | Technique | Sampling strategy |
|---|---|---|---|---|---|---|---|
| Houmsou et al. (2016) | 358 | 63 | *S. haematobium* | Nigeria | West Africa | Urine filtration | Simple random |
| Mafiana et al. (2003) | 209 | 150 | *S. haematobium* | Nigeria | West Africa | Urine filtration | All eligible children |
| Opara et al. (2007) | 126 | 25 | *S. haematobium* | Nigeria | West Africa | Urine filtration, dipsticks | All eligible children |
| Salawu and Alexander (2013) | 419 | 41 | *S. haematobium* | Nigeria | West Africa | Urine filtration, reagent strips | Random selection |
| Alemu et al. (2015) | 400 | 1 | *S. mansoni* | Ethiopia | East Africa | Kato Katz | Not specified |
| Alemu et al. (2016) | 401 | 45 | *S. mansoni* | Ethiopia | East Africa | Kato Katz | Two-stage cluster sampling |
| Lewetegn et al. (2019) | 214 | 9 | *S. mansoni* | Ethiopia | East Africa | Kato Katz | Not mentioned |
| Betson et al. (2010) | 1295 | 352 | *S. mansoni* | Uganda | East Africa | ELISA, Kato Katz | Not mentioned |
| Pinot de Moira et al. (2013) | 426 | 179 | *S. mansoni* | Uganda | East Africa | Kato Katz | Not mentioned |
| G/hiwot et al. (2014) | 374 | 33 | *S. mansoni* | Ethiopia | East Africa | SAF, Kato Katz | Systematic |
| Mueller et al. (2019) | 71 | 39 | *S. mansoni* | Tanzania | East Africa | POC CCA, Kato Katz | All eligible for participation |
| Nalugwa et al. (2015) | 3058 | 1203 | *S. mansoni* | Uganda | East Africa | Kato-Katz | Random cluster sampling |
| Ndokeji et al. (2016) | 71 | 43 | *S. mansoni* | Tanzania | East Africa | Kato-Katz | Random sampling |
| Niyituma et al. (2017) | 248 | 42 | *S. mansoni* | Rwanda | East Africa | Kato Katz, POC CCA | Random sampling |
| Odogwu et al. (2006) | 136 | 9 | *S. mansoni* | Uganda | East Africa | Kato Katz, POC CCA, SAF | Systematic random |
| Ruganuza et al. (2015) | 383 | 170 | *S. mansoni* | Tanzania | East Africa | Kato Katz, POC CCA | Systematic sampling |
| Rujeni et al. (2019) | 211 | 20 | *S. mansoni* | Rwanda | East Africa | Kato Katz, POC CCA | Convenient sampling |
| Sakari et al. (2017) | 361 | 18 | *S. mansoni* | Kenya | East Africa | Kato Katz, SAF | Random sampling |
| Stothard et al. (2011a) [1] | 247 | 19 | *S. mansoni* | Uganda | East Africa | Kato Katz, POC CCA, ELISA | Not mentioned |
| Stothard et al. (2011b) [2] | 242 | 115 | *S. mansoni* | Uganda | East Africa | Kato Katz | All willing to participation |
| Verani et al. (2011) | 216 | 79 | *S. mansoni* | Kenya | East Africa | Kato Katz, POC CCA, ELISA | Not mentioned |
| Masaku et al. (2020) | 653 | 77 | *S. mansoni* | Kenya | East Africa | Kato-Katz | Simple random |
| Sassa et al. (2020) | 305 | 11 | *S. mansoni* | Kenya | East Africa | Kato Katz, POC CCA | Simple random |
| Kemal et al. (2019) | 236 | 59 | *S. mansoni* | Ethiopia | East Africa | Kato-Katz | Systematic |
| Sousa-Figueiredo et al. (2010) | 363 | 225 | *S. mansoni* | Uganda | East Africa | Kato Katz, SEA-ELISA, CCA | Random sampling |

According to these authors, this reduction could have resulted from MDA programmes that had been on-going and focusing on SAC [10, 11]. We think that the prevalence estimates we obtained may be explained by the reduced geographic coverage for MDA campaigns and difficulties associated with accessing praziquantel in certain endemic settings. Also, the design of MDA campaigns has focused on SAC excluding other age groups such as PreSAC [17]. Furthermore, our results are based on specific study sites survey data which may not be representative of the entire countries where the studies were done. Our results further show that the risk of infection in Central and Southern Africa are relatively lower compared to Eastern and Western Africa. The paucity of studies focusing on schistosomiasis among PreSAC in Central Africa may have led to the low estimated PPE. No studies focusing on PreSAC were obtained from countries like Botswana, Lesotho, Namibia, and Zambia in Southern Africa, thus reinforcing the fact that PreSAC are generally excluded from schistosomiasis control initiatives.

The high prevalence estimates we observed may also be due to delayed detection of infection among PreSAC despite evidence that infections can occur in children as young as 6 months old [59]. A study conducted by Albonico et al. [75] suggested that many PreSAC may carry infection for several years until they reach the school-going age and become enrolled in

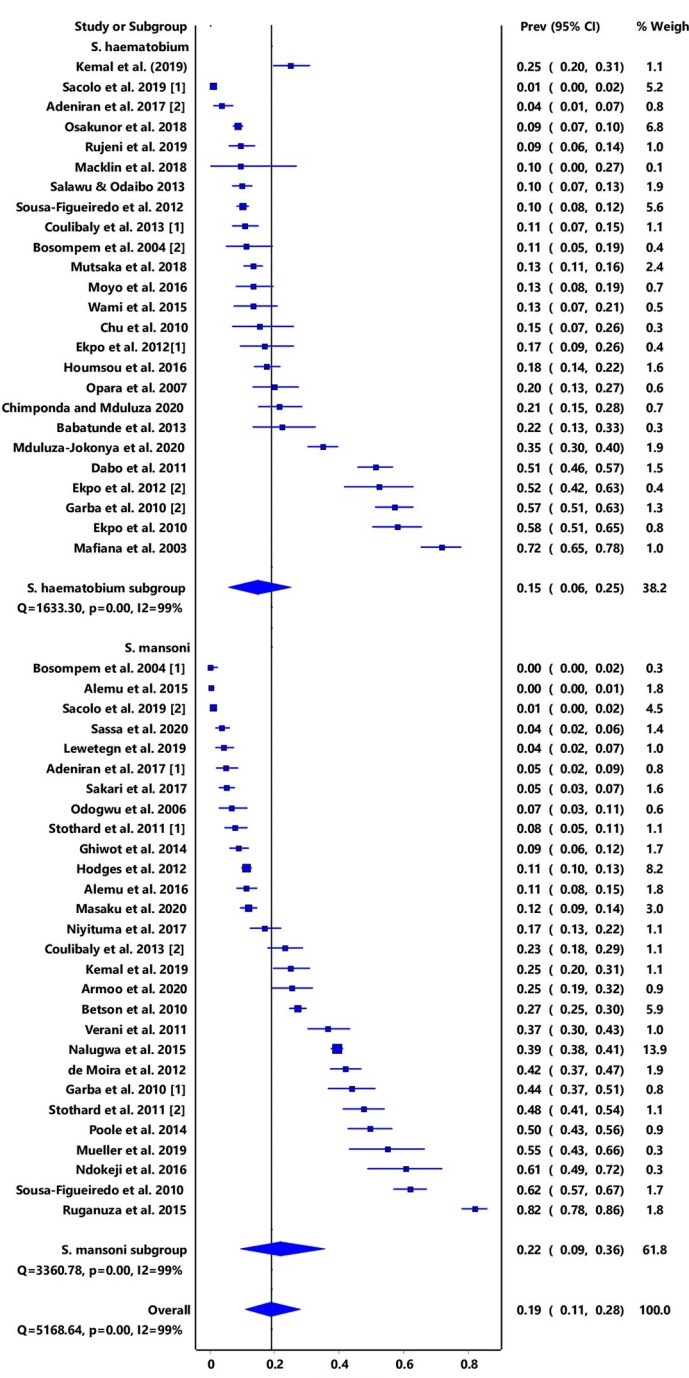

**Fig 2. Forest plot of subgroup PPE analysis for schistosome species.**

school. This is because disease monitoring and surveillance are more intense in the SAC group [12, 13]. Furthermore, disease control activities are often integrated into school-health programmes that take advantage of existing infrastructure in form of schools for cost-effectiveness in implementation of activities [14]. An extensive review conducted by Osakunor et al. [72] argued that schistosomiasis infections may perpetuate and go undetected or unnoticed in

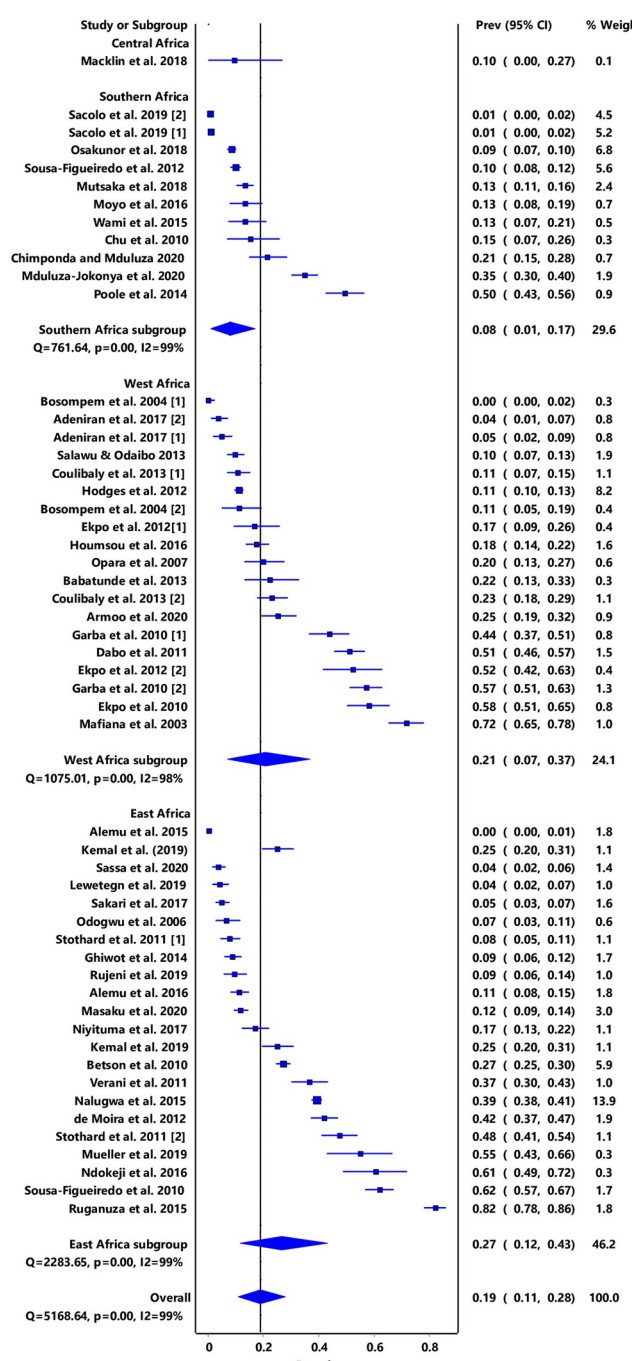

**Fig 3. Forest plot of subgrouped PPE analysis for infection prevalence in regions within sub-Saharan Africa.**

PreSAC due to operational difficulties and insufficient knowledge about the predisposing factors of infection in this age group. Also, Sacolo-Gwebu et al. [76], Mwai et al. [77] and Anguza et al. [78] found that poor knowledge, attitudes and misconceptions about schistosomiasis among caregivers increased the risks of infection among PreSAC. In addition, Mduluza and Mutapi [18] suggested that clinical misdiagnosis of infection owing to similarities in the

clinical manifestation between schistosomiasis and other infections may explain the high disease prevalence in PreSAC.

We caution that the prevalence estimates we have reported in this paper be interpreted as indicators and not absolute values. This is because most of the studies used the Kato-Katz technique and filtration methods for the detection of infection. Detection of infection in many poor settings has heavily relied on the Kato-Katz technique for *S. mansoni* and *S. japonicum* [79, 80] and filtration methods for *S. haematobium* [81]. These methods have low sensitivity, thus affecting the detection of infection in children whose infection intensity may be low [79, 80, 82, 83] and that tends to under-estimate positive cases [84, 85]. The development of new infection diagnostic tools such as urine-based POC-CCA cassette test, FLOTAC technique and serological detection based on anti-schistosome antibodies [81] may increase diagnostic accuracy even among PreSAC [37, 86, 87]. Furthermore, Sousa-Figueiredo et al. [68] observed that poverty and underfunding of health systems characterise communities where schistosomiasis infection among PreSAC thrives. Lack of basic diagnostic equipment for infection diagnosis has also been observed despite having competent health workers [73]. Exclusion of PreSAC in treatment campaigns may result in growth retardation and impaired cognitive development for some of the children [5, 6, 88], and other complications like prostate cancer that manifest in adulthood [18, 89].

If schistosomiasis control programmes are to achieve the goal of disease elimination [12, 90], there is a need to expand geographical coverage and improve the delivery of chemotherapy to all age groups. Various efficacy studies [70, 91, 92] have shown the safety and benefits of praziquantel in PreSAC. Furthermore, there is a need to encourage member countries to adopt the WHO recommendation of treating PreSAC [93] once the paediatric formulation of praziquantel which is under development becomes available. Therefore, to achieve total coverage of this group in MDA and schistosomiasis control activities, establishing the prevalence of infection and risks of infection and its associated factors is essential. In addition, the collaboration of various programmes implementing preventive chemotherapy at different scales and to the same/different players and scaling up of these activities will be essential in meeting the set targets and improving perceptions about schistosomiasis [9]. The levels of infection observed in the current study indicates the need to go beyond preventive measures to integrated control that also includes health education, provision of clean water and improvement of sanitation and snail control.

## Conclusion

The results obtained in our study are of clinical and public health importance as they confirm schistosome infection among PreSAC and highlight the need for anthelminthic treatment for this age group. If countries are to achieve the goal of eliminating schistosomiasis as a public health problem as set out in the NTD road map, there is a need to include PreSAC in schistosomiasis treatment programmes.

## Supporting information

**S1 File. Full electronic boolean search strategy used to identify studies with all search terms and limits for at least one database and the dates on which the database was accessed to obtain the data.**
(DOCX)

**S2 File. A quality assessment tool.**
(DOCX)

**S1 Fig. Doi plot the double arcsine transformed prevalence estimate of schistosomiasis among PreSAC in SSA (LFK index: 1.89).**
(TIF)

**S2 Fig. Funnel plot of the double arcsine transformed prevalence estimates of schistosomiasis among PreSAC in SSA.**
(TIF)

**S1 Data.**
(XLSX)

**S1 Checklist. PRISMA 2009 checklist.**
(DOC)

## Acknowledgments

The authors are indebted to Dr. Hlengiwe Sacolo-Gwebu for her pioneering work on schistosomiasis among preschool aged children (PreSAC) in Unkhanyakunde, Northern KwaZulu-Natal province. South Africa. This study contributes to current works focusing on understanding schistosomiasis among preschool aged children (PreSAC) in sub-Saharan Africa. We also acknowledge the precious inputs from the editors and anonymous reviewers who helped improve the content and quality of the paper.

## Author Contributions

**Conceptualization:** Chester Kalinda, Moses John Chimbari.

**Data curation:** Tafadzwa Mindu.

**Formal analysis:** Chester Kalinda.

**Methodology:** Chester Kalinda, Tafadzwa Mindu.

**Writing – original draft:** Chester Kalinda.

**Writing – review & editing:** Tafadzwa Mindu, Moses John Chimbari.

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
