## [Decision Letter · Decision Letter 0]

18 Nov 2020

PONE-D-20-21535

A systematic review and meta-analysis of schistosomiasis burden in pre-school aged children aged 5 years and below in sub-Saharan Africa for the period 2000 – 2020

PLOS ONE

Dear Dr. Kalinda,

Thank you for submitting your manuscript to PLOS ONE. After careful consideration, we feel that it has merit but does not fully meet PLOS ONE’s publication criteria as it currently stands. Therefore, we invite you to submit a revised version of the manuscript that addresses the points raised during the review process.

The systematic review by Kalinda and cols focuses in a age group that is a fragile part of the society mostly in underdeveloped Countries and constitutes a fraction very dependent on public policies for health and protection. The study has relevance and worth publication. However, there are still points that should be improved in a revised version of the MS as follows:

1-Please review and format the MS according to the ; 2-Include the pros and cons of the methods used by the cited papers that could influence or lead to mistaken conclusions. 3-Please follow all the suggestions/recommendations given by the 2 reviewers.

We look forward to receiving your revised manuscript.

Kind regards,

Marcello Otake Sato, Ph.D., D.V.M.

Academic Editor

PLOS ONE

Journal Requirements:

Additional Editor Comments (if provided):

The systematic review by Kalinda and cols focuses in a age group that is a fragile part of the society mostly in underdeveloped Countries and constitutes a fraction very dependent on public policies for health and protection. The study has relevance and worth publication. However, there are still points that should be improved as some correlations and validity of comparison to be better explained if the authors opt to revise the MS.

Reviewers' comments:

Reviewer's Responses to Questions

**Comments to the Author**

1. Is the manuscript technically sound, and do the data support the conclusions?

Reviewer #1: Yes

Reviewer #2: Yes

2. Has the statistical analysis been performed appropriately and rigorously? 

Reviewer #1: Yes

Reviewer #2: I Don't Know

3. Have the authors made all data underlying the findings in their manuscript fully available?

Reviewer #1: Yes

Reviewer #2: Yes

4. Is the manuscript presented in an intelligible fashion and written in standard English?

Reviewer #1: Yes

Reviewer #2: Yes

5. Review Comments to the Author

Reviewer #1: The study presented has value especially from the public health perspective of a neglected age group, the pre-school aged children, for Mass Drug Administration. Although the results presented here are just indicators, with the authors acknowledging the paucity of data, they nonetheless are still relevant to direct changes in policies towards diagnostic tools to use and the implementation of MDA programs.

I believe the paper can still be improved if the following are addressed.

Major comments:

Lines 71-72: Please clarify the basis for the Jan 1, 2000 – May 8, 2020 reckoning period.

Lines 130-137: Please clarify if the studies mentioned conducted the diagnostic tests on PSAC at random or did the subjects exhibit clinical manifestations prior to testing. In other words, if possible, can you indicate in the table the sampling method for selecting the PSAC to be tested?

Lines 130-137: Could you also break down in terms of percentages and mention here the different diagnostic tests used? This could help qualify your statement on the discussion part on the possible higher infection rates in PSAC due to the low sensitivity of some of the tests, particularly Kato-Katz.

Line 144: Provide relevant values for Central Africa for comparison with East Africa.

Line 163: Please indicate the prevalence estimates for both species based on your results.

Line 176-177: Do the countries mentioned herein have MDA programs against schistosomiasis? If so, is it possible that these countries did not report including PSAC but could have done so in the past?

Table 1: Under the column ‘Helminth,’ please write the scientific name properly. Write ‘S. mansoni’ and ‘S. haematobium’ instead.

Minor comments:

Line 18: Remove ‘,’ after ‘including’

Line 19: Change ‘Schistosome’ to ‘schistosome’

Line 19: Change ‘specie’ to ‘species’

Line 46: Remove ‘ ” ’ afger ‘risks’

Line 58: Change ‘from school children’ to ‘SACs’

Line 59: Add ‘and’ after ‘(PSAC),’

Line 61: Change ‘increase s’ to ‘increases’

Line 66: Does ‘SSA’ mean ‘Sub-Saharan Africa?’ If so, spell it out as it is mentioned for the first time here.

Line 80: Remove ‘,’ after ‘otherwise’

Line 98: Add ‘,’ after ‘heterogeneity’

Line 153: Add ‘,’ after ‘groups’

Line 167: Change ‘result’ to ‘resulted’

Line 177: Add ‘,’ after ‘Zambia’

Line 183: Add ‘in’ after ‘intense’

Line 199: Add ‘,’ after ‘sensitivity’

Table 1: Babatunde et al. Change ‘west Africa to ‘West Africa’

Reviewer #2: This manuscript presents interesting data on prevalence of Schistosoma in pre-school age children based on a systematic review and meta-analysis.

This study has been conducted well and enforced the fact of high prevalence of Schistosoma in pre-school age children in Africa. However, this provided findings were well known and I could not find any scientific novelties.

There are a few points to be checked.

Line 27, What is "LFK"? Abbreviation should not be used in this point.

Line 43 to 46, What are "Target 3.3", "Target 3.8" and "Target 3b"? It may be difficult to understand for readers unfamiliar to this scientific field.

Line 62, Please correct "This increase s health" to "This increases health".

Line 75, Please add double quotation marks to sub-Saharan Africa.

Line 163, Author mentioned "prevalence estimates for S. haematobium were higher than for S. mansoni". Is it correct? Please check it.

In Table 1, I think published year, sample size and prevalence of each paper should be added for easily understanding of tendency of Schistosoma prevalence .

6. PLOS authors have the option to publish the peer review history of their article (what does this mean?). If published, this will include your full peer review and any attached files.

Reviewer #1: No

Reviewer #2: No

---

## [Author Response · Author response to Decision Letter 0]

12 Dec 2020

Faculty of Agriculture and Natural Resources

Katima Mulilo Campus

Katima Mulilo

Namibia

12th December, 2020

Dear Editor-in-Chief, 

RE: Rebuttal Letter (PONE-D-20-21535)

We are grateful for the comments we received from the reviewers and we have considered all of them in revising our manuscript. We have presented our rebuttal in the table below showing how we have dealt with the reviewers’ comments.

Sincerely,

Chester Kalinda

Email: ckalinda@gmail.com, ckalinda@unam.na

---

## [Editor Report · Decision Letter 1]

15 Dec 2020

A systematic review and meta-analysis quantifying schistosomiasis infection burden in pre-school aged children (PreSAC) in sub-Saharan Africa for the period 2000 – 2020

PONE-D-20-21535R1

Dear Dr. Kalinda,

We’re pleased to inform you that your manuscript has been judged scientifically suitable for publication and will be formally accepted for publication once it meets all outstanding technical requirements.

Kind regards,

Marcello Otake Sato, Ph.D., D.V.M.

Academic Editor

PLOS ONE

Additional Editor Comments (optional):

All the suggested modifications were done.

---

## [Editor Report · Acceptance letter]

17 Dec 2020

PONE-D-20-21535R1 

A systematic review and meta-analysis quantifying schistosomiasis infection burden in pre-school aged children (PreSAC) in sub-Saharan Africa for the period 2000 – 2020 

Dear Dr. Kalinda:

I'm pleased to inform you that your manuscript has been deemed suitable for publication in PLOS ONE. Congratulations! Your manuscript is now with our production department. 

Kind regards, 

on behalf of

Dr. Marcello Otake Sato 

Academic Editor

PLOS ONE